# First Evidence of a Combination of Terpinen-4-ol and α-Terpineol as a Promising Tool against ESKAPE Pathogens

**DOI:** 10.3390/molecules27217472

**Published:** 2022-11-02

**Authors:** Bianca Johansen, Raphaël E. Duval, Jean-Christophe Sergere

**Affiliations:** 1SETUBIO SA, Naturopole, F-03800 Saint Bonnet de Rochefort, France; 2Université de Lorraine, CNRS, L2CM, F-54000 Nancy, France; 3Laboratoires Théa, F-63000 Clermont-Ferrand, France

**Keywords:** tea tree oil, terpinen-4-ol, α-terpineol, antibacterial activity, synergy, cytotoxicity, persisters

## Abstract

Antimicrobial resistance is a major public health issue raising growing concern in the face of dwindling response options. It is therefore urgent to find new anti-infective molecules enabling us to fight effectively against ever more numerous bacterial infections caused by ever more antibiotic-resistant bacteria. In this quest for new antibacterials, essential oils (or compounds extracted from essential oils) appear to be a promising therapeutic option. In the present work, we investigate the potential antibacterial synergy between a combination of terpinen-4-ol and α-terpineol (10:1) compared to standard tea tree oil. The minimum inhibitory concentration (MIC) and the minimum bactericidal concentration (MBC) were determined. Then, time kill assays, in vitro cytotoxicity and bactericidal activity on latent bacteria (persisters) were investigated. Finally, an in silico study of the pharmacokinetic parameters of α-terpineol was also performed. Altogether, our data demonstrate that the combination of terpinen-4-ol and α-terpineol might be a precious weapon to address ESKAPE pathogens.

## 1. Introduction

In 2017, the World Health Organization (WHO) published a list of bacteria divided into three categories according to the urgent need for new antibiotics against these pathogens [1]. Among them, unsurprisingly, were *Enterococcus faecium*, *Staphylococcus aureus*, *Klebsiella pneumoniae*, *Acinetobacter baumannii*, *Pseudomonas aeruginosa*, and *Enterobacter* spp. (i.e., ESKAPE) [2]. New antibiotics are therefore urgently needed to combat ESKAPE pathogens. Along with antibiotics, essential oils could be promising weapons among the different strategies currently being studied to fight those bacteria [3].

*Melaleuca alternifolia* essential oil (tea tree oil, TTO) and its compounds have been extensively studied for their biological properties, especially against bacteria [4,5,6]. In addition to its beneficial effects, TTO might also be irritating or toxic. We previously described a process for the fractionation of TTO, to separate toxic molecules from beneficial ones [7]. Interestingly, antimicrobial activity was maintained between TTO and the detoxified extract named Titroleane™.

Two major compounds of Titroleane™ are terpinen-4-ol and α-terpineol. Terpinen-4-ol is the main component of TTO, and is widely used in cosmetic and pharmaceutical products. [8] This monocyclic monoterpene alcohol has a broad antibacterial spectrum against both Gram-positive and Gram-negative bacteria [9,10,11]. It has been clearly demonstrated that terpinen-4-ol is not simply bacteriostatic, but is also bactericidal [12,13,14,15]. Terpinen-4-ol is also effective against antibiotic-resistant bacteria, including *Staphylococcus aureus* resistant to methicillin (i.e., MRSA), mupirocin, fusidic acid, vancomycin and linezolid [12,16,17]. Terpinen-4-ol has been shown to act in synergy with cefazolin, oxacillin, and meropenem, but not with gentamicin and vancomycin [18]. Experiments demonstrated that terpinen-4-ol might destroy the cell wall [19,20], and might affect protein and DNA synthesis [13]. Other experiments on *Burkholderia cepacia* support the hypothesis of membrane disruption, on both stationary-phase and log-phase cultures [17]. Interplay between the MexAB-OprM and MexCD-OprJ pumps in *Pseudomonas aeruginosa* may contribute to induce tolerance to terpinen-4-ol [21]. Terpinen-4-ol also has the ability to inhibit the biofilm formation of Gram-positive and Gram-negative bacteria [18,22]. Moreover, terpinen-4-ol inhibits adherence in biofilms by decreasing the expression of the *gbpA* (*Streptococcus mutans)* and *slpA* genes (*Lactobacillus acidophilus)* [23]. In a model using *Pseudomonas aeruginosa*, terpinen-4-ol inhibits quorum sensing (QS) by docking on QS receptors, thus downregulating all essential QS genes [24]. Moreover, terpinen-4-ol inhibits *Bacillus cereus* biofilm formation and spore germination, and reduces extracellular matrix synthesis, especially exopolysaccharides; it inhibits swarming motility and protease activity [25].

α-terpineol is another monocyclic terpene alcohol, found in essential oils such as *Citrus aurantium* (aka neroli), *Melaleuca alternifolia* or *Origanum vulgare*, widely used in cosmetics, soaps fragrances and pharmaceutical industry [26]. α-terpineol is well documented for its bactericidal activity on both Gram-positive and Gram-negative bacteria [15,27,28,29,30], in addition to other properties including antioxidant, anti-inflammatory, antiproliferative, and analgesic properties [31]. Remarkably, the molecule is also efficient against antibiotic-resistant bacteria [32]. Moreover, α-terpineol inhibits biofilm formation by destabilizing the cell membrane [19,33,34,35]. Investigations on the combination of several terpene alcohols demonstrated that α-terpineol was bactericidal against *S. typhimurium*, *Escherichia coli* O157:H7 and *Staphylococcus aureus* in a synergistic way when associated with linalool, but not when associated with eucalyptol, as well as the association of linalool with eucalyptol [36]. In the same study, the authors also showed by scanning electron microscopy that α-terpineol caused permeability alteration of the outer membrane, alteration of cell membrane function and leakage of intracellular materials.

As mentioned above, we have previously demonstrated that our detoxified extract from tea tree oil (i.e., Titroleane™) maintained the same antibacterial activity as the original product, on a wide range of bacteria, including antibiotic-resistant ones [7]. Moreover, as terpinen-4-ol and α-terpineol are the two main compounds retrieved in Titroleane™, we then decided to focus on the antibacterial properties of a combination of these two molecules.

## 2. Results

### 2.1. Antibacterial Activity of α-Terpineol and Terpinen-4-ol, Alone

It was previously shown that a commercial fraction of TTO named Titroleane™ exhibits a similar antimicrobial activity to TTO, although it was depleted from its monoterpenes which were expected to be the most important antimicrobial molecules in TTO [7]. We investigated the role of some major compounds of Titroleane™ to maintain the antibacterial activity.

Antibacterial activity of the two terpenic alcohols terpinen-4-ol and α-terpineol alone were first investigated on four Gram-positive and Gram-negative bacteria (Table 1). The Gram-positive *Staphylococcus aureus* sensitivity ranged between 1.25% and 2.50% with terpinen-4-ol and between 0.62% and 2.50% with α-terpineol. Both molecules exhibited identical bacteriostatic activities of 1.25% on MRSA. α-terpineol was slightly more efficient than terpinen-4-ol on the Gram-negative *Escherichia coli*, with respective MICs of 0.15–1.25% for α-terpineol versus 0.31–2.50% for terpinen-4-ol. *Pseudomonas aeruginosa* was less sensitive to both molecules in our assays, with respective MICs of 2.5% for terpinen-4-ol, and 1.25%–>2.5% for α-terpineol.

Table 1 summarizes results of MICs for terpinen-4-ol and α-terpineol in % (v:v), mM and mg/mL. As percentage as a concentration unit is commonly used for essential oils and their compounds, this unit is used in the rest of the article [7,15,17,18,37].

### 2.2. Synergstic Antibacterial Activity between α-Terpineol and Terpinen-4-ol

A potential synergy between terpinen-4-ol and α-terpineol was then investigated, using the checkerboard method [38]. Combination assays of terpinen-4-ol and α-terpineol were performed three times independently, on the same bacterial strains. Respective FICs were reported on Figure 1.

FIC and FICI values of α-terpineol and terpinen-4-ol from three independent assays on *Staphylococcus aureus*, methicillin-resistant *Staphylococcus aureus*, and *Escherichia coli* clearly showed a synergistic antibacterial effect of the two molecules for all combinations tested (Table 2). The same synergy was also observed for two out of three assays with *Pseudomonas aeruginosa*.

### 2.3. Comparison of the Bacteriostactic Activities of the α-Terpineol and Terpinen-4-ol Mixture and Tea Tree Oil

There is no clear ratio emerging for an optimal antibacterial activity on all four strains tested. According to the standard composition of TTO [39], terpinen-4-ol ranges from 35 to 48% of the global composition (m:m), and α-terpineol ranges from 2 to 5%. Hence, the ratio between terpinen-4-ol and α-terpineol is close to 10:1 (m:m). We thus designed a mixture of terpinen-4-ol and α-terpineol named Synterpicine™, with the same ratios of 10:1 (m:m) of terpinen-4-ol and α-terpineol. Interestingly, this 10:1 ratio is also similar to what was observed in the composition of Titroleane™ [7] with terpinen-4-ol ranging from 71 to 75% of the global composition (m:m), and α-terpineol ranging from 5 to 9%.

Antibacterial activity of Synterpicine™ was compared to standard TTO against the same bacterial strains (Table 3).

*Staphylococcus aureus* sensitivity ranged between 1.25% and 2.50% with Synterpicine™ and was 1.25% for TTO. MICs were, respectively, of 1.25% and 1.25–2.50% for TTO and Synterpicine™ against methicillin-resistant *Staphylococcus aureus* (MRSA). Considering the Gram-negative bacteria, Synterpicine™ was slightly more efficient than TTO on *Escherichia coli*, with respective MICs of 0.31–0.62% for Synterpicine™ versus 0.62–2.50% for TTO, while *Pseudomonas aeruginosa* had similar sensitivity (one serial dilution in one assay) for TTO and Synterpicine™, with respective MICs of 0.62–2.5% for TTO, and 1.25–2.50% for Synterpicine™.

### 2.4. Antibacterial Activity of α-Terpineol and Terpinen-4-ol on ESKAPE Pathogens

Synterpicine™ antibacterial properties were then investigated against the major pathogens of the ESKAPE group (*Enterococcus faecium*, *Staphylococcus aureus*, *Klebsiella pneumoniae*, *Acinetobacter baumannii* and *Enterobacter* spp.) [2]. *Escherichia coli* was included in the study.

Results of three independent MIC assays are summarized in Table 4.

MICs of Synterpicine™ were, respectively, 0.62%, 0.62–1.25% and 0.62–2.50% for *Klebsiella pneumoniae*, *Acinetobacter baumannii*, and *Enterobacter cloacae*. *Staphylococcus aureus* and *Pseudomonas aeruginosa* sensitivity to Synterpicine™ ranged between 1.25% and 2.50%. MRSA and *Enterococcus faecium* were slightly less sensitive to Synterpicine™ with respective MICs of 1.25–>2.50 and 2.50–>2.50. The MIC of *Escherichia coli* to Synterpicine™ ranged between 0.31% and 0.62%.

### 2.5. Bactericidal Activity of Synterpicine™

In addition to MICs, in order to better understand the effects of Synterpicine™ on some bacteria of major interest in health, minimum bactericidal concentrations (MBCs) were studied (Table 5).

In our MBC assays, *Klebsiella oxytoca* and *Klebsiella pneumoniae* appeared to be the more sensitive bacteria, with MBCs of 0.31–0.62% and 0.62%, respectively. The MBC of Synterpicine™ on *Escherichia coli* was 0.65–2.50%. *Enterobacter cloacae*, *Pseudomonas aeruginosa* and *Acinetobacter baumannii* had a similar sensitivity to Synterpicine™, with MBCs of 1.25–2.50%. In our experiments, we were not able to define the MBCs of *Staphylococcus aureus* (MIC = 2.50–>2.50%), MRSA (MIC > 2.50%) and *Enterococcus faecium* (MIC > 2.50%).

### 2.6. Time Kill Assays

In order to evaluate the effects of Synterpicine™ on the growth kinetics of bacteria, time kill assays were performed on *Staphylococcus aureus*, MRSA, *Escherichia coli* and *Pseudomonas aeruginosa*. Bactericidal activity is defined as a drop of more than 99.9% (3 Log10) of viable bacteria compared to the inoculum [40].

At a 4×MIC, Synterpicine™ had bactericidal activity against *Staphylococcus aureus* and MRSA in less than 1 h (Figure 2b,d). As a control, the antibiotic of reference ampicillin was efficient against *Staphylococcus aureus*, but only reached the bactericidal limit of a 3 Log10 decrease in 24 h (Figure 2a). As expected, MRSA could grow in the presence of ampicillin, though with less efficiency than in the absence of ampicillin (Figure 2c). *Escherichia coli* was sensitive to Synterpicine™, with a decrease of more than 3 Log10 in less than one hour (Figure 2f). With the antibiotic amoxicillin as a control, a decrease of more than 3 Log10 was observed only after 4 h (Figure 2e). *Pseudomonas aeruginosa* was sensitive to Synterpicine™, with a decrease of more than 3 Log10 in less than one hour (Figure 2h). With the antibiotic gentamicin as a control, a reduction of more than 3 Log10 was achieved only after 2 h (Figure 2g).

### 2.7. Bactericidal Activity of Synterpicine™ on Dormant Bacteria

Exponentially growing bacteria and dormant bacteria were compared for their sensitivity to Synterpicine™, using *Escherichia coli* as a model. DMSO (carrier) and amoxicillin (antibiotic) were used as negative and positive controls, respectively. All experiments were performed three times independently. Results are reported in Table 6.

Growth controls were performed on dormant versus exponentially growing *Escherichia coli*. After 24 h, exponentially growing bacteria increased by more than 3 Log10 (5.45 ± 0.05 Log10 CFU/mL at the beginning of the experiment versus 8.88 ± 0.05 Log10 CFU/mL after 24 h), while dormant bacteria numeration was almost unchanged (5.32 ± 1.37 Log10 CFU/mL at the beginning of the experiment versus 5.45 ± 0.65 Log10 CFU/mL after 24 h). This confirmed that the design of the experiment made it possible to study exponentially growing bacteria compared to dormant/non-growing bacteria in MBC experiments.

*Escherichia coli* was not sensitive to DMSO in the range of concentrations used (up to 2.50%), whatever its growing state. Exponentially growing bacteria were sensitive to amoxicillin at a MBC of 3.12–12.5 µg/mL, while non growing/latent bacteria were not sensitive to the antibiotic up to 25 µg/mL. Synterpicine™ MBC was 0.31% on exponentially growing *Escherichia coli*, and 0.62–1.25% on non-growing/dormant bacteria.

### 2.8. Cytotoxicity of Synterpicine™

Synterpicine™ cytotoxicity was evaluated in vitro on human foreskin fibroblasts (HFF) as previously described [7], and compared to TTO containing a similar ratio of terpinen-4-ol and α-terpineol (10:1). All experiments were performed with 0.30% DMSO, and the latter was used as a negative control of cytotoxicity. SDS (1%) was used as a positive control of cytotoxicity. After 24 h, no Synterpicine™ cytotoxicity was observed on HFF at a concentration of 0.025% (Figure 3). In the same conditions, TTO was slightly cytotoxic (i.e., viability = 81%) at a concentration of 0.006%.

### 2.9. In Silico Study of Pharmacokinetic Parameters

In silico analysis of the pharmacokinetic parameters of terpinen-4-ol has already been conducted [18], using the online Program SwissADME [41]. The same online Program was used to evaluate the theoretical pharmacokinetic parameters of α-terpineol (Table 7).

The molecular mass of α-terpineol is 154.25 g/mol, and is in accordance with Lipinski’s rule. However, this value is well below the optimal range for Ghose (180 ≤ MW ≤ 480), so the molecule violates this parameter.

Veber presents two essential parameters for molecules to be administrated orally: TPSA (Topological Polar Surface Area) and number of rotatable connections. α-terpineol has a TPSA of 20.23 Å^2^, and one rotatable connection. Therefore, the molecule has a high prospect of being used orally. α-terpineol has one hydrogen acceptor and one hydrogen donor, and is in accordance with Lipinski’s rule and has an excellent theoretical oral bioavailability. The Egan method and bioavailability score also support a good theoretical oral bioavailability.

The partition coefficient between n-octanol and water (Log Po/w) is commonly used to evaluate lipophilicity and the ability to cross plasma membranes. The Log Po/w consensus is 2.58 for α-terpineol. It meets the rules according to Lipinski (Log Po ≤ 5), Ghose (Log Po/w ≤ 5.6) and Egan (Log Po ≤ 5.6). It slightly differs from terpinen-4-ol consensus (2.60). This difference is due to one of the five predictive models, i.e., XLOGP3, an atomistic method based on the fragmental system of Wildman and Crippen [42].

The Log S—determined by the Ali method, ESOL model and SILICOS-IT—reflects the water solubility, which is key for the absorption and distribution of a molecule in the body. Log S for α-terpineol with these methods is, respectively, −2.78, −3.36 and −1.91, indicating that α-terpineol is soluble in water.

The SwissADME program also estimates other pharmacokinetic data. According to the online Program, gastrointestinal (GI) absorption is high, and the molecule can cross the blood–brain barrier (BBB). The molecule is not a P-glycoprotein (P-gp) substrate, and is not an inhibitor of Cytochrome P450 (CYP1A2, CYP2C19, CYP2C9, CYP2D6, CYP3A4).

## 3. Discussion

In a previous work, we physically separated TTO into two fractions, the first one containing monoterpenes, and a second one named Titroleane™ containing sesquiterpenes and terpene alcohols. The latter fraction is composed of more than twenty molecules, the most abundant ones having been already described [7]. In the present study, we have focused on major monoterpene alcohols identified in Titroleane™ -, i.e., terpinen-4-ol and α-terpineol, which are well used in the cosmetic and pharmaceutical industries [8,26].

In a first experiment, we evaluated the antibacterial activities of terpinen-4-ol and α-terpineol alone, using standard microdilution assays according to the Clinical and Laboratory Standards Institute guidelines [43]. Using *Staphylococcus aureus*, MRSA, *Escherichia coli*, and *Pseudomonas aeruginosa* as bacterial targets, we confirmed that terpinen-4-ol and α-terpineol had MICs in accordance with previously described experiments [9,10,12,13,15,16,18,22,29,30,33,36,44,45].

Combining essential oils appears to be a promising strategy for antimicrobial treatments [46]. The combination of essential oils or their compounds can lead to synergistic, antagonisms, or additive antimicrobial effects (reviewed in [47,48]). Extreme antagonism between essential oils or their compounds has been described [49,50], highlighting the need to carefully look at potential interactions when combining them. Synergies have been described for TTO, against *Streptococcus mutans* when mixed with *Citrus limon* and *Piper nigrum* essential oils [51], *Streptococcus agalactiae* when combined with *Lavandula officinalis*, and *Staphylococcus aureus* when associated to *Origanum vulgare* [37]. Functional interactions between some TTO compounds have also been studied by TS Yang et al. [52]. These authors investigated the potential synergies/antagonisms between linalool, terpinen-4-ol, α-terpineol and p-cymene, the major active compounds of *Glossogyne tenuifolia*, also present in TTO in significant amounts. Combinations were tested on *E. coli* O157:H7, *Salmonella enterica*, *Vibrio parahaemolyticus*, *Staphylococcus aureus*, *Listeria monocytogenes*, *Streptococcus mutans*, and *Streptococcus sanguinis*, using the checkerboard analysis. Linalool was tested in combination with terpinen-4-ol and α-terpineol, and p-cymene with linalool, terpinen-4-ol and α-terpineol. No synergy or antagonism was observed in these experiments, but intriguingly, the terpinen-4-ol/α-terpineol combination has not been tested. One explanation should be that terpinen-4-ol and α-terpineol are two isomers of terpineol, and the authors considered these molecules possessed similar biological activities and might only present additive effects. Nevertheless, for the first time, the present work clearly establishes the synergistic antibacterial activity of terpinen-4-ol and α-terpineol combination against six major pathogenic bacteria responsible for thousand infections every year. Such a synergy suggests that in addition to common mode of action of the two molecules increased membrane permeability and enhanced cell wall and cell membrane integrity [19], additional specific process (ses) might occur. This also indicates that the doses of these two molecules might be decreased when associated. The consequence reduced side effects such as toxicity, allergenicity, possible resistance adaptation, or even fragrance while maintaining the antibacterial properties.

In TTO, as well as in its previously described fraction Titroleane™, terpinen-4-ol and α-terpineol concentrations are 35–48%:2–5% and 71–75%:5–9%, respectively [7]. Thus, the terpinen-4-ol:α-terpineol ratios in TTO and Titroleane™ are closed to 10:1. We have chosen to focus on a mixture of terpinen-4-ol and α-terpineol with a ratio of 10:1, which we named Synterpicine™. MICs of TTO and Synterpicine™ performed on *Staphylococcus aureus*, MRSA, *Escherichia coli* and *Pseudomonas aeruginosa* were similar, suggesting that the mixture of terpinen-4-ol and α-terpineol in a 10:1 ratio was sufficient to mimic TTO antibacterial activity. Since TTO also contains other molecules with antibacterial activity [4,53], this suggest that some complex interactions such as antagonisms between compounds probably occur inside TTO, as previously proposed [53].

In 2017, the World Health Organization (WHO) published a list of bacteria with three categories depending on the urgency to develop new antibiotics against these pathogens (the most critical ones being *Enterococcus faecium*, *Staphylococcus aureus*, *Klebsiella pneumoniae*, *Acinetobacter baumannii*, *Pseudomonas aeruginosa*, and *Enterobacter* spp. (acronym: ESKAPE) [54]. MICs results of Synterpicine™ on ESKAPE pathogens an MRSA clearly highlight the potential benefits of the mixture to prevent or even treat major infections.

MBC studies clearly indicate that the Synterpicine™ mixture is not only bacteriostatic, but also bactericidal, as previously shown for terpinen-4-ol and α-terpineol alone [13,14,16,17,18,27,34,35,55]. Results show that the MBCs are very close to the MIC, diverging from a maximum of one serial dilution. Hence, the respective MBCs and MICs of Synterpicine™ are identical (*Klebsiella pneumoniae*, *Pseudomonas aeruginosa*) or differ from a maximum of one serial dilution (*Staphylococcus aureus*, MRSA, *Enterococcus faecium*, *Staphylococcus aureus*, *Acinetobacter baumannii*, and *Enterobacter cloacae*). Similar results have been observed previously for terpinene-4-ol alone, with a MIC/MBC ratio of 1:2 on *Staphylococcus strains* [18,56].

Bactericidal activity is defined as a decrease of more than 1000 times (3 Log10) of viable bacteria compared to the inoculum [40]. Time kill assays confirm the bactericidal properties of Synterpicine™ on *Staphylococcus aureus, Escherichia coli*, MRSA and *Pseudomonas aeruginosa* in less than 1 h at 4×MICs, which is faster than ampicillin on *Staphylococcus aureus*, amoxicillin on *Escherichia coli* and gentamicin on *Pseudomonas aeruginosa* in standard conditions. These results are in accordance with previously published results with terpinen-4-ol [16,56] and α-terpineol [35] alone. Time kill assays thus support standard MBC assays, and show that Synterpicine™ is an efficient bactericide against *Staphylococcus aureus*, MRSA, *Escherichia coli* and *Pseudomonas aeruginosa*, making this mixture of terpinen-4-ol and α-terpineol a good antibacterial natural candidate.

Some bacteria can adapt to a dormant form under stresses such as nutrient starvation, antibiotic exposure, acid, or oxidative stress [57,58]. This subpopulation of dormant cells, also called persisters, can adopt a metabolic inactivity and become stress-tolerant (review in [59]). In addition, it has been suggested that chronic infections were probably caused by revivified persister bacteria [60,61], thus highlighting the importance of developing antibacterial molecules efficient against these dormant cells. Consequently, the observed bactericidal effect of Synterpicine™ on *Escherichia coli* persisters makes the mixture a promising candidate to fight chronic infections. Further experiments are needed to confirm the antipersister activity on other bacterial species, and extend applications of Synterpicine™.

The cytotoxicity of terpinen-4-ol and α-terpineol has been investigated for a long time, mostly on immortalized cell lines such as breast, lung, pancreatic, prostate, colorectal, gastric and leukemia cell lines [62,63]. Interestingly, it appeared that cancer cells were more sensitive to terpinen-4-ol than primary cells [64,65], as was observed with TTO [66]. Thus, terpinen-4-ol and α-terpineol might be good candidates as anticancer drugs [62,63,67,68]. TTO as well as its components terpinen-4-ol and α-terpineol were evaluated as Generally Recognized As Safe (GRAS) [69]. Though, α-terpineol has been described to exhibit antiproliferative effects when associated to other molecules [70]. We thus performed cytotoxicity assays on human primary cells (HFF) to ensure that Synterpicine™ was kept under TTO cytotoxic levels. Our experiments clearly show that Synterpicine™ is less cytotoxic than TTO. Interestingly, measured survival of the cells was similar with 0.05% Synterpicine™ (83%) and 0.006% TTO (81%), suggesting that Synterpicine™ is ten times less cytotoxic than TTO.

Terpinen-4-ol and α-terpineol are two isomers sharing the same chemical formula C_10_H_18_O. Thus, they are supposed to share very close pharmacokinetic parameters. In silico analysis of the pharmacokinetic parameters of terpinen-4-ol has already been performed [18], using the online Program SwissADME [41], and the present results obtained for α-terpineol confirmed this hypothesis. Results are similar for both isomers, only differing in one of the five methods, i.e., XLOGP3, which is an atomistic method based on the fragmental system of Wildman and Crippen [42] to evaluate lipophilicity and ability to cross plasma membranes.

The partition coefficient between n-octanol and water (Log Po/w) is commonly used to evaluate lipophilicity and ability to cross plasma membranes. The Log Po/w consensus is 2.58 for α-terpineol. It meets the rules according to Lipinski (Log Po ≤ 5), Ghose (Log Po/w ≤ 5.6) and Egan (Log Po ≤ 5.6). Interestingly, it slightly differs from the terpinen-4-ol consensus (2.60). This difference is due to one of the five predictive models, i.e., XLOGP3, an atomistic method based on the fragmental system of Wildman and Crippen [42]. In summary, α-terpineol and terpinen-4-ol share very close in silico pharmacokinetic parameters. The pharmacokinetic profile suggests good theoretical oral bioavailability, adequate absorption and distribution of α-terpineol in vivo. In silico pharmacokinetic results should be confirmed in vivo. Anyway, results from in silico analysis support the potential of α-terpineol as a drug candidate, alone or in association with terpinen-4-ol.

## 4. Materials and Methods

### 4.1. Chemicals

#### 4.1.1. Active Compounds

Terpinen-4-ol (CAS 562-74-3) and α-terpineol (CAS 98-55-5) were purchased from Acros Organics (Fisher Scientific, Villebon-sur-Yvette, France). Standard tea tree essential oil (TTO), according to the norm ISO/FDIS 4730:2017, was purchased from Helpac (Auzon, France).

#### 4.1.2. Samples Preparation

To enhance the solubility of TTO, terpinen-4-ol and α-terpineol, dimethyl sulfoxide (DMSO) (Sigma Aldrich, Saint-Quentin-Fallavier, France) was used as a solvent. Oils were diluted at 10% final volume (v:v) in 10% DMSO. All experiments were performed simultaneously with a control solution of 10% DMSO.

Synterpicine™ composition is 75% terpinen-4-ol and 7.5% α-terpineol in water (v:v).

### 4.2. Antimicrobial Activities

#### 4.2.1. Bacteria

*Klebsiella oxytoca* (ATCC 49131) and *Klebsiella pneumoniae* (ATCC 700603) were purchased from the American Type Culture Collection (ATCC).

*Staphylococcus aureus* (CIP 4.83), methicillin-resistant *Staphylococcus aureus* MRSA (CIP 107422), *Escherichia coli* (CIP 53.126), *Acinetobacter baumannii* (CIP 70.34), *Pseudomonas aeruginosa* (CIP 82.118), *Enterococcus cloacae* (CIP 103475), and *Enterococcus faecium* (CIP 102379) were purchased from “Collection de l’Institut Pasteur” (CIP, Pasteur Institute Collection).

Master stocks of bacterial strains were stored in a −80 °C freezer.

#### 4.2.2. Growth Conditions and Inoculum Preparation

All strains were cultivated on trypticase soy agar and broth (236920 and 211825, BD, France) at 37 °C, except *Acinetobacter baumannii*, which was grown at 22 °C, as previously described [7]. The bacteria were grown on the plate and then a single colony was transferred to a second plate. The inoculum was prepared from isolates of the second agar plate in TPS solution (Tryptone 0.1% (211705, BD, Le Pont-de-Claix, France) and sodium chloride 0.85% (S9888, Sigma Aldrich, Saint-Quentin-Fallavier, France)), and was then diluted in adequate broth at 5 × 10^5^–1 × 10^6^ CFU/mL.

#### 4.2.3. The Minimum Inhibitory Concentration (MIC) and the Minimum Bactericidal Concentration (MBC) Determination

MIC values were measured using the microdilution broth method, according to the Clinical and Laboratory Standards Institute guidelines [43], with modification on the broths used to fit with organism requirements for growth [7]. Briefly, 96-well microplates were prepared with three controls—growth control without product, negative control without bacteria, and solvent control (DMSO from 2.50% to 0.0049%). Ampicillin (10 µg/mL), amoxicillin (25 µg/mL) and gentamycin (25 µg/mL) were used as positive controls. Wells were prepared by serial dilutions from 2.50% to 0.0049% with 50 µL of product and 50 µL of suspension; microplates were then incubated in growth conditions defined above, then MICs were read visually. A volume of 50 µL of each well with no visible growth was plated on Tryptone Soja agar plates, then incubated for 24 h before counting. The MBC was defined as the lowest concentration without growth after 24 h. A twofold variation between two MICs, corresponding to one dilution, was considered as non-significant. For persister studies, the same protocol was used except that bacteria were inoculated in PBS 3 h before the experiment to slow the growth and reach stationary growth due to the lack of nutrients. Counting was performed 3 h after inoculation (inoculum) and 24 h post-assay (MBC).

#### 4.2.4. Latent Bacteria (Dormant, Persister Cells)

Latent bacteria were obtained by a successive combination of two previously described protocols [71,72]. Mueller–Hinton broth was depleted from its essential nutrients by successive cultures of *Escherichia coli* for 48 h. After each cycle, the culture was centrifuged at 7500× *g*. The supernatant was filtered on a 0.22 µm filter, then re-inoculated, until no growth could be observed. The obtained depleted MH broth was then inoculated with an inoculum of *Escherichia coli* at 5 × 10^5^–1 × 10^6^ CFU/mL to determine the standard MBC, as described above.

#### 4.2.5. Synergy

Synergy evaluation between terpinen-4-ol and α-terpineol was performed using the checkerboard method as previously described [38] using the standard MIC protocol described above, and serial dilutions of order 2. The assays were performed on *Staphylococcus aureus*, MRSA, *Escherichia coli* and *Pseudomonas aeruginosa*. MH broth was used as negative control. Inoculum alone was used as positive control. In order to assess the interactions between two compounds *A* and *B*, fractional inhibitory concentrations (FICs) and the fractional inhibitory concentration index (*FIC*) are calculated. The FICs of each of the *A* and *B* compounds are calculated as follows:$$FIC\;A=\frac{MIC\;\left(AB\right)}{MIC\;A}\;$$
$$FIC\;B=\frac{MIC\;\left(AB\right)}{MIC\;B}\;$$
where *MIC* (*AB*) is the *MIC* of the combination of *A* and *B*, *MIC A* is the *MIC* of compound *A*, and *MIC B* is the *MIC* of compound *B*. The fractional inhibitory concentration index (FICI) is the sum of *FIC A* and *FIC B*. A FICI of ≤0.5 indicates synergism, between 0.5 and 1 indicates additive effects, between 1 and 2 is considered as indifference, and ≥2 is antagonism [73].

#### 4.2.6. Time Kill

Time kill experiments were adapted from [74,75]. Compounds were tested at concentrations corresponding to the MIC, the 2×MIC and the 4×MIC. DMSO was tested in parallel, at the same concentrations than the terpinen-4-ol and α-terpineol mix. Antibiotic concentrations for MRSA were identical to *S. aureus*. Bactericidal experiments were performed in round-bottomed 96-well plates. Bacteria were inoculated at 5 × 10^5^ CFU/mL in 100 µL final volume of MH broth. Each test was performed in triplicates, then pooled for counting on TS agar broth.

### 4.3. In Vitro Cytotoxicity

Human foreskin fibroblast cell lines were purchased from ATCC collection SCRC-1041. Cells were grown in MEM (10370-047, Gibco, Paisley, UK) with the addition of 10% heat-inactivated fetal bovine serum (10270, Gibco, Paisley, UK), 2 mM of L-glutamine (P04-80100, Pan biotech, Aidenbach, Germany), 100 µg/mL ampicillin (A9518, Sigma-Aldrich, Madrid, Spain), 0.10 mU/mL of penicillin–streptomycin (P4333, Sigma-Aldrich, Burlington, MA, USA) and 2.50 µg/mL of amphotericin B (P06-01050, PAN biotech, Aidenbach, Germany). Cells were maintained at 37 °C in a 5% CO_2_ humidified atmosphere. The cytotoxicity was carried out by Neutral Red coloration assay for investigating cell viability [76]. Cells were seeded on 96-well plates, at a concentration of 10^5^–10^6^ cells per mL in completed MEM, and incubated for 48 h before the addition of the sample to be tested. Various concentrations of samples, all prepared in 0.3% DMSO (final concentration in wells v:v) and sonicated at room temperature for 10 min at 45 kHz (Transsonic TIH5, ELMA, Düsseldorf, Germany) before use, were then added to wells with a final volume of 200 µL. Negative (no treatment), solvent (DMSO 0.30%), and positive (0.10% SDS, BP166, Fisher) controls were included. Each condition was tested three times per assay. Exposure periods of 24 h were chosen for determining and comparing the in vitro cytotoxicity potential of samples. After incubation, the supernatant was removed and cells were washed three times with PBS (10010-015, Gibco, Paisley, UK) before adding Neutral Red (229810250, ACROS organics, Mumbai, India) and solution prepared in completed MEM at 0.005% (v:v). The plate was then incubated for an additional 3 h. The Neutral Red solution was washed three times in PBS, the coloration was solubilized in acetic acid 1%:ethanol 50% (v:v) and the absorbance was measured at 540 nm using a microplate reader (Infinite M200 pro, Tecan, Crailsheim, Germany). The cell survival rate expressed in percentage was determined by comparing the absorbance values obtained with treated cells and with DMSO.

### 4.4. Statistical Analysis

The data were analyzed using a one-way ANOVA Dunnet statistical analysis, which allows for the comparison of data to a control—here the solvent control [77]. Statistical analysis was carried out with the free online SAS version (SAS Studio version 9.4), with the use of Oracle VM VirtualBox version 5.1, and VMware Player version 5.0 (SAS University Edition).

### 4.5. In Silico Pharmacokinetic Parameters

In silico analysis of the pharmacokinetic parameters of α-terpineol was performed using the free online SwissADME Program (www.swissadme.ch, (accessed on 5 August 2020) Swiss Institute of Bioinformatics^®^, Lausanne, Switzerland). This program evaluates lipophilicity, water solubility, and druglikeness. It makes it possible to predict whether a drug candidate violates the rules of Lipinski [78], Ghose [79], Veber [80] and Egan [81]. Lipophilicity is classically estimated by evaluating the partition coefficient between n-octanol and water (Log p/o). Five predictive models are used in SwissADME: XLOGP3 [82], WLOGP [42], MLOGP [83,84], SILICOS-IT and iLOGP. Water solubility is estimated using the two topological methods derived from the ESOL model [85] and from Ali et al. [86]. A third predictor for solubility developed by SILICOS-IT is also used by SwissADME. The skin permeability coefficient (Kp) is a multiple linear regression model adapted from Potts and Guy [87] which aims at predicting skin permeation, based on molecular size and lipophilicity. The BOILED-Egg model allows the prediction of blood–brain barrier (BBB) permeation and passive gastrointestinal (GI) absorption [88]. The machine learning algorithms developed for SwissADME also allow to evaluate if a molecule can be a substrate for permeability glycoprotein (P-gp), and the five major isoforms of Cytochrome P450 (CYP1A2, CYPC19, CYP2C9, CYP2D6, and CYP3A4) [41]. The results are binary: either Yes (substrate) or No (not a substrate). Drug-likeness is evaluated according to the Lipinski [78], Ghose [79], Veber [80], and Egan [81] methods, and the Abbot bioavailability score [89].

## 5. Conclusions

We demonstrated that terpinen-4-ol and α-terpineol act in synergy to kill bacterial pathogens. Particularly, the 10:1 ratio between terpinen-4-ol and α-terpineol—that we named Synterpicine™—has bactericidal effects on ESKAPE pathogens, and on the dormant bacteria. In vitro, Synterpicine™ is also less cytotoxic than standard TTO, a GRAS plant extract. In silico pharmacokinetic parameters suggest that Synterpicine™ is water soluble, can be used orally, gastrointestinal absorption is high, can cross the plasma membranes and the blood–brain barrier. 

Taken together, the in vitro and in silico data make the combination of terpinen-4-ol and α-terpineol a very promising candidate to help us to fight bacterial pathogens, including ESKAPE ones. Confirmation will come from in vivo experiments.

## 6. Patents

Patent application was submitted (application FR2012343).

## Figures and Tables

**Figure 1 molecules-27-07472-f001:**
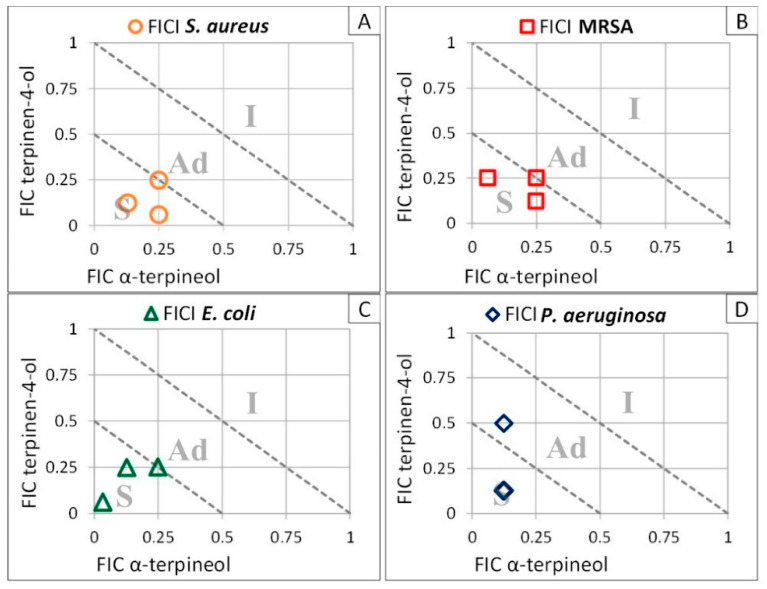
Synergy between α-terpineol and terpinen-4-ol. Synergy assays between the two terpenic alcohols, α-terpineol and terpinen-4-ol, were performed on: (**A**) *Staphylococcus aureus* ○, (**B**). methicillin-resistant *Staphylococcus aureus* □, (**C**) *Escherichia coli* ∆, and (**D**) *Pseudomonas aeruginosa* ◊. The range of concentrations tested is 0.039–2.50% for terpinen-4-ol, and 0.001–2.50% for α-terpineol (v:v). Values were directly visible on the graphics. S: synergy; Ad: additive; I: indifference; FIC: fractional inhibitory concentration; FICI: fractional inhibitory concentration index (*n* = 3).

**Figure 2 molecules-27-07472-f002:**
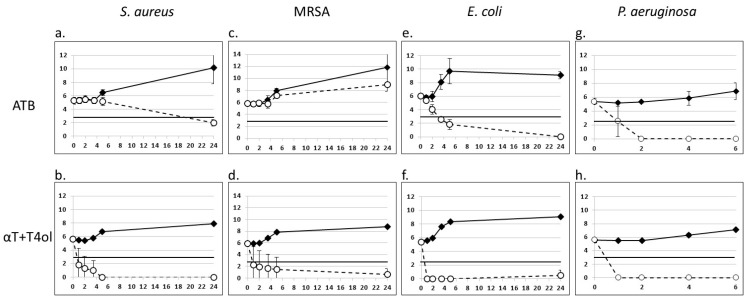
Time kill assays of Synterpicine™. Bacteria were incubated in the presence of (**a**,**c**): ampicillin; (**e**): amoxicillin; (**g**): gentamycin; (**b**,**d**,**f**,**h**): Synterpicine™. Black diamond: control; white circle: 4×MIC. The horizontal black line symbolizes the limit for a bactericide activity (>3 Log10 decrease). ATB: antibiotic (reference); αT + T4ol: Synterpicine™. *x*-axis: time (hours); *y*-axis: Log10 versus CFU/mL (*n* = 2).

**Figure 3 molecules-27-07472-f003:**
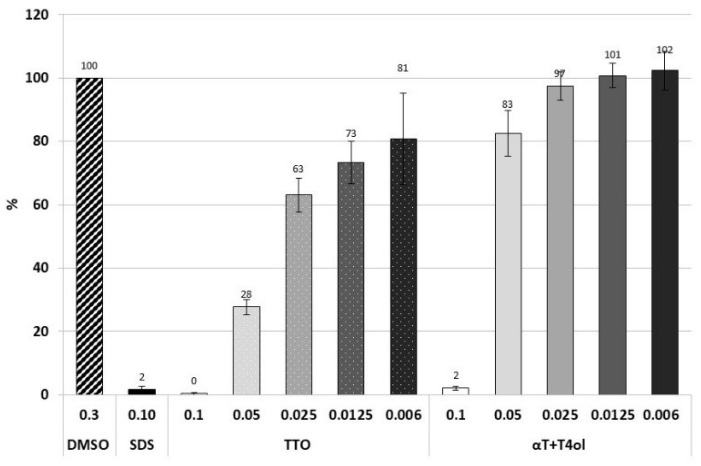
In vitro cytotoxicity of Synterpicine™ versus TTO. Survival percent of TTO and Synterpicine™ (αT + T4ol) on human foreskin fibroblasts were reported. Samples were prepared in 0.30% DMSO. Concentrations ranged from 0.1% to 0.006% (v:v). SDS 0.10% was used as a positive control of cellular cytotoxicity. DMSO 0.3% alone was used as a negative control. The experiment was performed three times in triplicates on different days.

**Table 1 molecules-27-07472-t001:** Minimum inhibitory concentration (MIC) of α-terpineol and terpinen-4-ol. MIC assays were performed on four Gram-positive and Gram-negative species. The range of concentrations tested is 0.00499–2.50% (v:v). MRSA: methicillin-resistant *Staphylococcus aureus*. Antibiotics used as controls are ampicillin for *Staphylococcus aureus* and MRSA, amoxicillin for *Escherichia coli*, and gentamycin for *Pseudomonas aeruginosa* (*n* = 3).

Species/Strains	Terpinen-4-ol			α-Terpineol			Antibiotic
	% (v:v)	mM	mg/mL	% (v:v)	mM	mg/mL	(µg/mL)
*Staphylococcus aureus*	1.25–2.50	74.6–149	11.5–23.0	0.62–2.50	37.3–149	5.76–23.0	0.039–0.078
MRSA	1.25	74.6	11.5	1.25	74.6	11.5	1.25–>2.50
*Escherichia coli*	0.31–2.50	18.6–149	2.88–23.0	0.15–1.25	9.33–74.6	1.44–11.5	1.56–6.50
*Pseudomonas aeruginosa*	2.50	149	23.0	1.25–>2.50	74.6–>149	11.5–>23.0	0.19–0.39

**Table 2 molecules-27-07472-t002:** Bactericidal synergy between α-terpineol and terpinen-4-ol. FIC values of α-terpineol and terpinen-4-ol were obtained from three independent assays on the four bacteria reported in Figure 1. The effect deduced from the calculated FICI are detailed. Antibiotics used as controls are ampicillin for *Staphylococcus aureus* and MRSA, amoxicillin for *Escherichia coli*, and gentamycin for *Pseudomonas aeruginosa.* FIC: fractional inhibitory concentration; FICI: fractional inhibitory concentration index.

Species/Strains	FIC Terpinen-4-ol	FIC α-Terpineol	FICI	Effect	Antibiotic (µg/mL)
*Staphylococcus aureus*	0.062	0.125	0.312	Synergy	5.00–10.00
	0.125	0.25	0.25	Synergy	
	0.25	0.25	0.5	Synergy	
MRSA	0.125	0.062	0.312	Synergy	1.25
	0.25	0.25	0.375	Synergy	
	0.25	0.25	0.5	Synergy	
*Escherichia coli*	0.062	0.031	0.093	Synergy	3.12–12.50
	0.25	0.125	0.375	Synergy	
	0.25	0.25	0.5	Synergy	
*Pseudomonas aeruginosa*	0.125	0.125	0.25	Synergy	0.19–4.56
	0.125	0.125	0.25	Synergy	
	0.5	0.125	0.625	Additive	

**Table 3 molecules-27-07472-t003:** Minimum inhibitory concentration (MIC) of TTO and Synterpicine™. MIC assays were performed on 4 Gram-positive and negative species. The range of concentrations tested is 0.00499–2.50% (v:v). TTO: tea tree oil; Synterp.: Synterpicine™; MRSA: methicillin-resistant *Staphylococcus aureus* (*n* = 3).

Order	Species/Strains	MIC	
		TTO	Synterp.
Bacillales	*Staphylococcus aureus*	1.25	1.25–2.50
	MRSA	1.25–2.50	1.25–>2.50
Enterobacteriales	*Escherichia coli*	0.62–2.50	0.31–0.62
Pseudomonadales	*Pseudomonas aeruginosa*	0.62–2.50	1.25–2.50

**Table 4 molecules-27-07472-t004:** Minimum inhibitory concentration (MIC) of Synterpicine™ on ESKAPE bacteria. MIC assays were performed on *Enterococcus faecium*, *Staphylococcus aureus*, MRSA, *Klebsiella pneumoniae*, *Acinetobacter baumannii, Enterobacter cloacae* and *Escherichia coli*. The range of concentrations tested is 0.00499–2.50% (v:v). Synterp.: Synterpicine™; MRSA: methicillin-resistant *Staphylococcus aureus* (*n* = 3).

Order	Species/Strains	MIC
		Synterp.
Bacillales	*Staphylococcus aureus*	1.25–2.50
	MRSA	1.25–>2.50
Lactobacillales	*Enterococcus faecium*	2.50–>2.50
Enterobacteriales	*Enterobacter cloacae*	0.62–2.50
	*Klebsiella pneumoniae*	0.62
	*Escherichia coli*	0.31–0.62
Pseudomonadales	*Acinetobacter baumannii*	0.62–1.25
	*Pseudomonas aeruginosa*	1.25–2.50

**Table 5 molecules-27-07472-t005:** Minimum bactericidal concentration (MBC) of Synterpicine™. MBC assays were performed on bacteria with already defined MICs. The range of concentrations tested is 0.00499–2.50% (v:v). Synterp.: Synterpicine™; MRSA: methicillin-resistant *Staphylococcus aureus* (*n* = 3).

Order	Species/Strains	MBC
		Synterp.
Bacillales	*Staphylococcus aureus*	>2.50
	MRSA	2.50–>2.50
Lactobacillales	*Enterococcus faecium*	>2.50
Enterobacteriales	*Escherichia coli*	0.62–2.50
	*Enterobacter cloacae*	1.25–2.50
	*Klebsiella oxytoca*	0.31–0.62
	*Klebsiella pneumoniae*	0.62
Pseudomonadales	*Acinetobacter baumannii*	1.25–2.50
	*Pseudomonas aeruginosa*	1.25–2.50

**Table 6 molecules-27-07472-t006:** MBC of Synterpicine™ on latent and exponentially growing *Escherichia coli*. Concentrations ranged from 0.048 to 25 µg/mL for amoxicillin, and from 0.0049% to 2.50% for Synterpicine™ and DMSO. Growth controls on exponential and latency/non-growing bacteria were performed at 0 (T0) and 24 h (T24). DMSO (carrier) and amoxicillin (antibiotic) were used as negative and positive controls, respectively (*n* = 3).

		Exponential	Latency
Bacterial count (Log10 CFU/mL)	T0	5.45 ± 0.05	5.32 ± 1.37
	T24	8.88 ± 0.005	5.45 ± 0.65
MBC	Synterpicine™ (%)	0.31	0.62–1.25
	DMSO (%)	>2.50	>2.50
	Amoxicillin (µg/mL)	3.12–12.50	>25

**Table 7 molecules-27-07472-t007:** In silico study of the pharmacokinetic parameters of α-terpineol. Parameters are interpreted as follows: Class (Ali): insoluble < −10 < poorly soluble < −6 < moderately soluble < −4 < soluble < −2 very soluble < 0 highly soluble; Log Po/w ≤ 5; Lipinski: MM ≤ 500, H-bond donors ≤ 5, H-bond acceptors ≤ 10; Ghose: 180 ≤ MM ≤ 480, 20 ≤ number of atoms ≤ 70, 40 ≤ molar refractometry ≤ 130, −0.4 ≤ Log Po/w ≤ 5.6—Veber: number of rotatable bonds ≤ 10, TPSA ≤ 140 Å^2^—Egan: Log Po/w ≤ 5.88, TPSA ≤ 131.6 Å^2^.

Physicochemical Properties	
Formula	C_10_H_18_O
Molecular weight	154.25 g/mol
num. heavy atoms	11
Num. arom. heavy atoms	0
Num. rotatable bonds	1
Num. H-bond acceptors	1
Num. H-bond donors	1
Molar refractivity	48.80
TPSA	20.23 Å^2^
**Lipophilicity**	
Log Po/w (iLOGP)	2.51
Log Po/w (XLOGP3)	3.39
Log Po/w (WLOGP)	2.50
Log Po/w (MLOGP)	2.30
Log Po/w (SILICOS-IT)	2.44
Consensus Log Po/w	2.58
**Water solubility**	
Log S (ESOL)	−2.78
Solubility	2.54 × 10^−1^ mg/mL; 1.64 × 10^−3^ mol/L
Class	Soluble
Log S (Ali)	−3.36
Solubility	6.75 × 10^−2^ mg/mL; 4.38 × 10^−4^ mol/L
Class	Soluble
Log S (SILICOS-IT)	−1.91
Solubility	1.92 mg/mL; 1.24 × 10^−2^ mol/L
Class (Ali)	Soluble
**Pharmacokinetics**	
GI absorption	High
BBB permeant	Yes
P-gp substrate	No
CYP1A2 inhibitor	No
CYPC19 inhibitor	No
CYP2C9 inhibitor	No
CYP2D6 inhibitor	No
CYP3A4 inhibitor	No
Log Kp (skin permeation)	−4.93 cm/s
**Druglikeness**	
Lipinski	Yes; 0 violation
Ghose	No; violation: MW < 160
Veber	Yes
Egan	Yes
Bioavailability score	0.55

## Data Availability

Not applicable.

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
