# Peer review of "First Evidence of a Combination of Terpinen-4-ol and α-Terpineol as a Promising Tool against ESKAPE Pathogens"

_molecules, 2022, doi:10.3390/molecules27217472_

Round 1
Reviewer 1 Report
In the present manuscript, the authors wish to report data demonstrating that the combination of terpinen-4-ol and α-terpineol might be a good combination to deter bacteria of the ESKAPE group.
To be acceptable, minor corrections could improve the quality of work.
1- For instance, a thoughtful revision of the English language and typewriting, and coherence of sentences.
2- It would be interesting to have the Bacteriostatic and bactericide concentrations of pure terpenoids in micrograms per mL and microMolar to be possible a comparison among treatments and conventional antimicrobial concentrations.
3- It is recommended the authors discuss why they performed a synergistic assay and in the final combination of terpenoids choose the ratio 10:1, based on the Synterpicine™ composition (75% terpinen-4-ol and 7.5% α-terpineol in water, v:v). What was the best composition based on the synergistic assay?
4- If the cytotoxic concentration on healthy cells is 0.05%, 10 times lower in comparison to TTO oil, still inferior to the bactericide concentration (between 0.65 - 2.5%). Based on these numbers, what do the authors think about whether the product is safe to use and developed into an antibacterial drug?
.
Author Response
Reviewer 1
In the present manuscript, the authors wish to report data demonstrating that the combination of terpinen-4-ol and α-terpineol might be a good combination to deter bacteria of the ESKAPE group.
To be acceptable, minor corrections could improve the quality of work.
1- For instance, a thoughtful revision of the English language and typewriting, and coherence of sentences.
The manuscript has been proofread by a native English speaker.
2- It would be interesting to have the Bacteriostatic and bactericide concentrations of pure terpenoids in micrograms per mL and microMolar to be possible a comparison among treatments and conventional antimicrobial concentrations.
Authors’ response
We are conscious that researchers dealing with conventional antibiotics are not aware of the use of percentage as a concentration unit. In fact, the use of percentage as a concentration unit is very common in the world of essential oils and their compounds study, especially terpenoids such as terpenic alcohols (here terpinen-4-ol and α-terpineol). This justifies the use of percentage as a concentration unit in Table 2 for example, where a mixture of terpinen-4-ol and α-Terpineol is compared to TTO.
Nevertheless, we have added conversions in other units in Table 1, so readers who wish to use other units can do it easily (conversions are expressed in mM and mg/mL).
3- It is recommended the authors discuss why they performed a synergistic assay and in the final combination of terpenoids choose the ratio 10:1, based on the Synterpicine™ composition (75% terpinen-4-ol and 7.5% α-terpineol in water, v:v). What was the best composition based on the synergistic assay?
Authors’ response
We thank reviewer 1 for this interesting question.
If one superimpose the 4 graphs of figure 1, there is no clear ratio emerging for an optimal antibacterial activity on all of the 4 strains. We previously showed that both TTO and Titroleane® who have antibacterial activities both had a terpinene-4-ol:terpineol close to 10:1. We thus choose to mimic the natural ratio between the two enantiomers for further analysis, and named this combo Synterpicine™.
These comments have been incremented in the text, Line 130.
4- If the cytotoxic concentration on healthy cells is 0.05%, 10 times lower in comparison to TTO oil, still inferior to the bactericide concentration (between 0.65 - 2.5%). Based on these numbers, what do the authors think about whether the product is safe to use and developed into an antibacterial drug?
Authors’ response
In vitro cytotoxicity assays are very sensitive assays. They do not reflect directly in vivo toxicity (concentrations cannot be transposed). For example, in cosmetics, irritability potential is evaluated by using in vitro cytotoxicity assays on corneal cells. The percentage of death cells at a fixed concentration do not mean that the same percentage of cells of the user (human) will be killed, hopefully! Here we use in vitro cytotoxicity assays on primary cells to compare TTO vs terpinen-4-ol + α-terpineol. Comparison of the cytotoxicity between several active compounds is very common in the literature, to evaluate the less toxic one (hit-to-lead experiments in drug development for example).
TTO is recognized as GRAS (Generally Recognized As Safe). According to the literature, there is no death due to TTO absorption, even with more than ten milliliters absorbed. Only reversible symptoms occur. TTO is already used by pharmaceutical companies to treat demodicosis (due to the mite parasite Demodex). Terpinen-4-ol and α-terpineol are also widely used in cosmetics, perfumes, and pharmaceutical industries (see lines 40-41 and 61-62 for references). Thus these 3 compounds are widely used at concentrations thousands of times more elevated than the toxic concentrations in in vitro assays that we and others reported. Thus, we believe that a combo of terpinen-4-ol and α-terpineol are a good candidate for an antimicrobial therapy.

Reviewer 2 Report
In this study, the minimum inhibitory and bactericidal concentrations of terpinen-4-ol,α-Terpineol and Synterpicine™ against a variety of bacteria were determined. The time killing assay, in vitro cytotoxicity and bactericidal activity of synterpicine™ were performed. The pharmacokinetic parameters of α-terpineol were analyzed using online software. In a number of previous studies, terpinen-4-ol and α-terpinedol has been confirmed to have a killing effect on a variety of bacteria, including Staphylococcus aureus and Pseudomonas aeruginosa. Many studies have also discussed their combined effect with antibiotics and the antibacterial mechanism. Many results of this study have been reported before. What is its main innovation?
comments and suggestions:
1. The title of this paper states that terpinen-4-ol and α-Terpineol
have antimicrobial activity on lESKAPE Pathogens, however, not every experiment in this paper targeted these six superbacteria ,the research content cannot support the title.
2. In the preface, the meaning of the thesis is not adequately expressed.
3. The concentration unit of antimicrobial activity in this article is expressed by percentage. Percentage is a relatively crude concentration expression method, which is inconsistent with the commonly used drug concentration units and cannot be compared with other antibacterial agents. It is recommended to convert the percentage to µg/mL or M.
4. Whether the strain used in the study is a standard strain or a clinical strain, the specific name should be marked out, not just the general name of the strain.
5. In the part of synergistic effect results, the FIC value calculated after synergistic effect is displayed in the main text, and it is suggested to display the combination results of different drug concentrations as the attached figure.
6. Line113-115, according to the FICs values of terpineen-4-ol and α-terpineol against Staphylococcus aureus provided by the author, FICIs calculation results should be 0.312, 0.25 and 0.75, while the author calculated results are 0.312, 0.25 and 0.5. Please check the data carefully. If FICIs is 0.75, greater than 0.5, the two cannot be considered to have synergistic effect.
7. Line137-144, The content in this paragraph is the same as that in Table 2. It is recommended to simplify.
8. Line137-150 Table 2 showed that the antibacterial activity of α-terpineol and terpineen-4-ol mixture was worse than that of TTO, but the results of Figure 1 (Line106-124) showed that α-terpineol and terpineen-4-ol had significant synergistic effect. The two experimental results were contradictory, how to explain this?
9. The results of the antibacterial activity test in the paper can be combined into one table, because many data in the table are duplicated, and combined together can better compare the antibacterial effect of single use and combined use.
10. The number of bacteria selected in each experiment was inconsistent. Four strains were detected in 2.1, 2.2, 2.3, 2.6. 8 strains were detected in 2.4, and 9 strains in 2.5. Only one strain was detected in 2.7. The experimental design is not reasonable enough.
11. Line 232 2.8 Figure 3 shows that the concentration of 0.025% Synterpicine™ had no cytotoxicity on HFF237, while the concentration of 0.05% Synterpicine™ had significant cytotoxicity. However, in the previous antibacterial activity experiments (Tables 2, 3, and 4), the effective antibacterial concentration of Synterpicine™ ranged from 0.31 to 2.5%, much higher than 0.025%, which means that synterpicine™ is very cytotoxic while exerting antibacterial effect. Is there a future for synterpicine™? Similarly, TTO was mildly toxic to cells at a concentration of 0.006%, while TTO exerted antibacterial activity at a concentration of at least 0.62% (Table 2). With such cytotoxicity, can they be used for antimicrobial therapy?
12. Line378-381 The author clearly states that synterpicine™ is less cytotoxic than TTO, but then states that synterpicine™ is 10 times more cytotoxic than TTO. Such inconsistent description, did the author think carefully to examine when writing the article?
13. This paper introduces the preparation of Escherichia coli in dormant state according to the references. How to prove that the prepared bacteria is in dormant state? How is it different from normal bacteria?
14. Pharmacokinetics was only analyzed by online procedure, and the results were all speculative. Further experimental verification is needed.
15. Line 431 In the description of MIC and MBC measurement methods, ampicillin, amoxicillin and gentamicin were used as positive drug controls, but in the results, the MIC and MBC values of positive drugs were not shown in Table 1-4.
16. Line 449 The method describes the preparation of Staphylococcus aureus dormant bacteria, but the results section displays dormant Escherichia coli(Line 207)?
17. There are too many references and it is recommended to simplify.

Author Response
Reviewer 2
In this study, the minimum inhibitory and bactericidal concentrations of terpinen-4-ol,α-Terpineol and Synterpicine™ against a variety of bacteria were determined. The time killing assay, in vitro cytotoxicity and bactericidal activity of synterpicine™ were performed. The pharmacokinetic parameters of α-terpineol were analyzed using online software. In a number of previous studies, terpinen-4-ol and α-terpinedol has been confirmed to have a killing effect on a variety of bacteria, including Staphylococcus aureus and Pseudomonas aeruginosa. Many studies have also discussed their combined effect with antibiotics and the antibacterial mechanism. Many results of this study have been reported before. What is its main innovation?
Authors’ response
We agree with you that many studies also discussed combined effects (including synergies). We discussed this point lines 299-318. But, to the best of our knowledge, this is the first time that a study demonstrates that terpinen-4-ol and α-Terpineol - two commonly used terpenoids from natural origin widely used in cosmetic, hygiene, perfumes and pharmaceutical industries – work synergistically to kill ESKAPE pathogens and dormant cells.
comments and suggestions:
1. The title of this paper states that terpinen-4-ol and α-Terpineol have antimicrobial activity on lESKAPE Pathogens, however, not every experiment in this paper targeted these six superbacteria, the research content cannot support the title.
Authors’ response
The work described in this article is composed of three successive parts. The first one is about the discovery that terpinen-4-ol and α-terpineol act synergistically on standard bacterial models to kill them (both Gram-positive and Gram-negative bacteria used in standard norms such as challenge tests and other antibacterial assays). The second part is about the translation of these initial models to ESKAPE pathogens, because of the major interest to identify new molecules to fight against these “super-bacteria” (which is our daily objective). The third part deals with a first series of in vitro and in silico analysis of a combo of terpinen-4-ol and α-terpineol get more detailed data (time kill studies), extension of the bactericidal properties to dormant cells which is not common in the community of antibacterial researchers, and to ensure there is no major toxicity or pharmacokinetics lock.
Thus, it appears to us that the core (and goal) of this work (identifying promising molecules to fight ESKAPE pathogens) has to appear in the title, since we clearly demonstrate the in vitro bacteriostatic and bactericidal properties of the combo of terpinen-4-ol : α-terpineol. We are aware that this work is not sufficient to claim that this combination of two molecules can be used in therapy, in patients. Long is the road to this claim (in vitro assays such to understand the mechanisms of action, resistance appearance, in vitro and in vivo (small animals) ADMET, clinical trials…), reason why we add the notion of “first evidence” in the title.
Moreover, minor additional comments have been added to the text to clarify this. Thus, we suggest to keep the title.
2. In the preface, the meaning of the thesis is not adequately expressed.
Authors’ response
We're really sorry, but we don't understand what the reviewer means. What does the reviewer mean by "preface"? or "the meaning of the thesis"?
3. The concentration unit of antimicrobial activity in this article is expressed by percentage. Percentage is a relatively crude concentration expression method, which is inconsistent with the commonly used drug concentration units and cannot be compared with other antibacterial agents. It is recommended to convert the percentage to µg/mL or M.
Authors’ response
We are conscious that researchers conducting study on “conventional” antibiotics are not familiar with the use of percentage as a concentration unit. On the contrary, the use of percentage as concentration unit is very common in the field of study of essential oils and their compounds, such as, but not limited to terpenoids such as terpenic alcohols (here terpinen-4-ol and α-terpineol). There are many examples of articles, including those published on the subject in Molecules (MDPI). It allows to compare different EO with different densities/MW. This justifies the use of percentage as a concentration unit in Table 2 for example, where a mixture of terpinen-4-ol and α-Terpineol is compared to TTO.
Nevertheless, we have added conversions in other units in Table 1, so readers who wish to use other units can do it easily (conversions are expressed in mM and mg/mL).
4. Whether the strain used in the study is a standard strain or a clinical strain, the specific name should be marked out, not just the general name of the strain.
Authors’ response
All strains are standard strains, bought from ATCC (American Type Culture Collection) and CIP (Collection de l’Institut Pasteur). Their references are detailed in paragraph 4.2.1. Bacteria. In order to simplify the reading of the manuscript, it did not seem necessary to us, to repeat each time the origin of the strains in the main text.
5. In the part of synergistic effect results, the FIC value calculated after synergistic effect is displayed in the main text, and it is suggested to display the combination results of different drug concentrations as the attached figure.
Authors’ response
A table has been added to Figure 1.
Comments in the text have been simplified.
6. Line113-115, according to the FICs values of terpineen-4-ol and α-terpineol against Staphylococcus aureus provided by the author, FICIs calculation results should be 0.312, 0.25 and 0.75, while the author calculated results are 0.312, 0.25 and 0.5. Please check the data carefully. If FICIs is 0.75, greater than 0.5, the two cannot be considered to have synergistic effect.
Authors’ response
Thank you for bringing us with this typing mistake. Going back to experimental data, it appears that one should read 0.25 instead of 0.5 for the third value of FIC Terpinen-4-ol. The FICI is correct.
Correction has been done, and it clearly appears in the new table (comment #5).
7. Line137-144, The content in this paragraph is the same as that in Table 2. It is recommended to simplify.
Authors’ response
Lines 137-144 are a description of Table 1, with some comments on the relative sensitivities of the strains.
We suggest to keep the text.
8. Line137-150, Table 2 showed that the antibacterial activity of α-terpineol and terpineen-4-ol mixture was worse than that of TTO, but the results of Figure 1 (Line106-124) showed that α-terpineol and terpineen-4-ol had significant synergistic effect. The two experimental results were contradictory, how to explain this?
Authors’ response
There are two points in your question.
The first is related to MIC repeatability. It is well known among microbiologists performing MIC assays that these MIC experiments are not perfectly repeatable. Results can often vary of one serial dilution due to the technique (serial dilutions of order 2). This is the reason why MIC experiments are (or they should be) performed three times, in three independent experiments (triplicates are not sufficient). The results in Table 2 illustrate this situation. As an example, considering Staphylococcus sensitivity, 3 independent experiments have been performed. With TTO, MICs were of 1.25% three times, while for Synterpicine, MICs were twice of 1.25 and one time of 2.5, giving the feeling that Synterpicine was less efficient than TTO, because of well-known inter-experiments variations.
The second point concerns the complexity of essential oils. It is now well documented that the tenths and even hundreds of molecules constituting an EO can either act independently, in addition, in synergy or as antagonists (see references). Thus, even if 2 molecules from an EO act in synergy for a biological activity, they can be less efficient than the EO that contains these two molecules, because of the other molecules present in the EO. There is no contradiction, the two experiments just investigate different questions.
9. The results of the antibacterial activity test in the paper can be combined into one table, because many data in the table are duplicated, and combined together can better compare the antibacterial effect of single use and combined use.
Authors’ response
We agree that it is possible to group some experimental results in one table. However, It appears to us that Tables and Figures must follow the different steps of the story : 1/ MIC activity of the 2 molecules alone, synergy between the 2 molecules, 3/ comparison of the bacteriostatic activity of the combo vs TTO, 4/ bacteriostatic activity on ESKAPE pathogens, 5/ not only bacteriostatic activity, but also bactericidal activity of on ESKAPE pathogens, 6/ kinetics (time kill) of bactericidal activity, 7/ bactericidal activity on dormant cells (in vitro cytotoxicity + in silico ADME to support the potential interest of this association). Thus, we believe that (i) for each point of the story, a specific table or figure is needed, and (ii) doing so, it greatly facilitates the reading of the manuscript.
10. The number of bacteria selected in each experiment was inconsistent. Four strains were detected in 2.1, 2.2, 2.3, 2.6. 8 strains were detected in 2.4, and 9 strains in 2.5. Only one strain was detected in 2.7. The experimental design is not reasonable enough.
Authors’ response
This manuscript describes the results obtained by Bianca Johansen during her PhD. This follows a story (question 1). To be exhaustive, we first screened several compounds of TTO independently and in association. A combination appeared to be of particular interest (Tables 1 and 2 and Figure 1). The logical way is to expend the study to major pathogens of interest (Tables 3 and 4). Additional work was performed on a part of the major models as a POC, to support these findings (Figure 2, Table 5). Please also note that not all of the strains can be studied in the same way: dormant cells are difficult to obtain routinely (see bibliography), and we were able to obtain E. coli persisters but not S. aureus for example. This is also something well known. Figure 3 and Table 6 are integrated to support the interest for the combination of terpinen-4-ol and α-Terpineol.
Thus, to us, the experimental design is correct and reasonable.
11. Line 232 2.8 Figure 3 shows that the concentration of 0.025% Synterpicine™ had no cytotoxicity on HFF237, while the concentration of 0.05% Synterpicine™ had significant cytotoxicity. However, in the previous antibacterial activity experiments (Tables 2, 3, and 4), the effective antibacterial concentration of Synterpicine™ ranged from 0.31 to 2.5%, much higher than 0.025%, which means that synterpicine™ is very cytotoxic while exerting antibacterial effect. Is there a future for synterpicine™? Similarly, TTO was mildly toxic to cells at a concentration of 0.006%, while TTO exerted antibacterial activity at a concentration of at least 0.62% (Table 2). With such cytotoxicity, can they be used for antimicrobial therapy?
Authors’ response
In vitro cytotoxicity assays are very sensitive assays. They do not reflect directly in vivo toxicity (concentrations cannot be transposed). For example, in cosmetics, irritability potential is evaluated by using in vitro cytotoxicity assays on corneal cells. The percentage of death cells at a fixed concentration do not mean that the same percentage of cells of the user (human) will be killed, hopefully! Here we use in vitro cytotoxicity assays on primary cells to compare TTO vs terpinen-4-ol + α-terpineol. Comparison of the cytotoxicity between several active compounds is very common in the literature, to evaluate the less toxic one (hit-to-lead experiments in drug development for example).
TTO is recognized as GRAS (Generally Recognized As Safe). According to the literature, there is no death due to TTO absorption, even with more than ten millilitres absorbed. Only reversible symptoms occur. TTO is already used by pharmaceutical companies to treat demodicosis (due to the mite parasite Demodex). Terpinen-4-ol and α-terpineol are also widely used in cosmetics, perfumes, and pharmaceutical industries (see lines 40-41 and 61-62 for references). Thus these 3 compounds are widely used at concentrations thousands of times more elevated than the toxic concentrations in in vitro assays that we and others reported. Thus, we believe that a combo of terpinen-4-ol and α-terpineol are a good candidate for an antimicrobial therapy.
12. Line378-381 The author clearly states that synterpicine™ is less cytotoxic than TTO, but then states that synterpicine™ is 10 times more cytotoxic than TTO. Such inconsistent description, did the author think carefully to examine when writing the article?
Authors’ response
We are sorry, but we disagree with reviewer 1. There must be a mistake. Indeed, as far as we can read, it is not written anywhere that Synterpicine is 10 times more cytotoxic than TTO in lines 378-381.
13. This paper introduces the preparation of Escherichia coli in dormant state according to the references. How to prove that the prepared bacteria is in dormant state? How is it different from normal bacteria?
Authors’ response
A characteristic of the dormant state is that bacteria do not grow, with a slow metabolism (see references 74 and 75 for example). The common way to prove that bacteria are in a dormant state during an experiment is to check the absence of growth (lanes 108 to 219 in the manuscript, and lane bacterial count / column latency for the counts at T0 and T24).
14. Pharmacokinetics was only analyzed by online procedure, and the results were all speculative. Further experimental verification is needed.
Authors’ response
Correction has been done in the text (Line 398).
15. Line 431 In the description of MIC and MBC measurement methods, ampicillin, amoxicillin and gentamicin were used as positive drug controls, but in the results, the MIC and MBC values of positive drugs were not shown in Table 1-4.
Authors’ response
Results for the antibiotics have been integrated into Tables 1b (MIC) and 2b (MBC) in the manuscript
16. Line 449 The method describes the preparation of Staphylococcus aureus dormant bacteria, but the results section displays dormant Escherichia coli (Line 207)?
Authors’ response
Reviewer 1 is right. There was an error in the method, one should read “Escherichia coli” instead of “Staphylococcus aureus”.
Correction has been done in the text.
17. There are too many references and it is recommended to simplify.
Authors’ response
This last remark is particularly surprising. This is the first time that a reviewer has criticized an excess of bibliographic references. Moreover, there is no recommendation on this subject (i.e. limiting the number of bibliographical references) in the instructions to authors.
There is no recent (or even old) review on terpinen-4-ol and α-terpineol. This might have eased the management of references in the article. Writing such reviews is something we are thinking about. Nevertheless, some references have been deleted in the new version when possible.

Round 2
Reviewer 2 Report
The article is not innovative enough, and the research method is relatively simple. Pharmacokinetics is only analyzed by online program, which needs further experimental verification.
Author Response
Reviewer 2
The article is not innovative enough, and the research method is relatively simple. Pharmacokinetics is only analyzed by online program, which needs further experimental verification.
Authors’ response
We sincerely thank reviewer 2 for his relevant comments which allow us to have a very good scientific discussion... However, we note that reviewer 2 does not comment on round R1 and the 17 responses we provided to his first comments...
“The article is not innovative enough”
We thank reviewer 2 for this interesting remark. Anyway, if we consider that our research work is the very first experimental work which is interested in a potential synergy between the Terpinen-4-ol and α-Terpineol; we honestly think we can call our research work “innovative”.
“and the research method is relatively simple”
Again, we thank reviewer 2 for this constructive remark. To determine the antimicrobial properties of compounds, whatever their chemical nature, the methods which can be used are not very numerous. On the contrary, they are limited in number, and must obey a number of recommendations issued by internationally recognized agencies.
All the methods used (MIC determination, CMB determination, synergy study, time kill assay, etc.) are all methods validated and recommended by the CLSI (Clinical and Laboratory Standards Institute). In addition, the bacterial strains used are all strains from national or international collections (CIP, ATCC)... So yes, it is possible, like reviewer 2, to think that these methods are "simple" but they are robust and are the basis of all scientific research in the field of research into new anti-infectives...
Please also note that the specific protocols developed for the study of dormant bacteria is not so simple (heavy work and controls for the validation of the technique).
“Pharmacokinetics is only analyzed by online program, which needs further experimental verification.”
Reviewer 2 is completely right, our work to obtain pharmacokinetic data has been performed using software. It is therefore "in silico" data. Indeed, these are preliminary data which will in any case have to be confirmed on animal models if we pursue the development of this association of two terpenes... which we have every intention of doing. We explicitly mention the need for further experimental verification in the article.
That being said, this software exists, and we think that we would be wrong not to use it, as other authors do elsewhere. It also allows to comment the results obtained previously with terpinene-4-ol (cited article published in 2020 in Int. J. Mol. Sci.) and alpha-terpineol (our study). Of note, at the end of this answer you will find a list of 10 publications (from Pubmed) which have used this software... and we are voluntarily limited to the last ten publications.
Furthermore, as scientific researchers concerned about the impact of their research work on Society, particularly in the field of animal experimentation, we are looking for alternative methods and information sharing as recommended. by the REACH program (https://echa.europa.eu/fr/animal-testing-under-reach).
